# Towards Understanding the Role of Humans in Collaborative Tasks

Ayesha Jena and Elin A. Topp
Department of Computer Science
Lund University, Sweden
[ayesha.jena,elin_a.topp]@cs.lth.se

## ABSTRACT

This paper explores the dynamics of human-robot collaboration through a comparative study of human-assisted and system-assisted approaches in a search and rescue application. Leveraging virtual environments and mixed-reality interfaces, the study evaluates task performance, workload, usability, and subjective experiences of participants. Results indicate that the system-assisted approach significantly improves task completion time and accuracy in identifying critical elements, and reduces perceived workload compared to human-assisted methods. Subjective assessments reveal valuable insights into user preferences and challenges, informing recommendations for system refinement and protocol development. Findings highlight the potential of human collaboration in enhancing operational effectiveness and promoting seamless collaboration between humans and robots in cluttered and high-risk environments. Interactions aimed at synchronizing goals, task states, and actions can be facilitated through virtual, augmented, and mixed-reality environments providing an intuitive platform for understanding interaction dynamics.

## CCS CONCEPTS

• **Human-centered computing** → **Empirical studies in interaction design**; **User studies**; **Collaborative interaction**; *User interface management systems*; **Gestural input**.

## KEYWORDS

human-robot interaction, collaborative robots, interaction design, human-assisted, system-assisted, search-and-rescue, virtual environments, communication modalities, user study

**ACM Reference Format:**
Ayesha Jena and Elin A. Topp. 2024. Towards Understanding the Role of Humans in Collaborative Tasks. In *7th International Workshop on Virtual, Augmented, and Mixed-Reality for Human-Robot Interactions at HRI 2024 (VAM-HRI '24)*. ACM, Boulder, CO, USA, 7 pages.

## 1 INTRODUCTION

In a time marked by rapid technological advancements, robotics continues to evolve and permeate various facets of society ranging from manufacturing and healthcare to disaster response and space

exploration. The synergy between humans and robots holds immense promise across these diverse domains as this paradigm shift transcends the traditional notion of robotics where there was little to no interaction between humans and robots. With the increased integration of robots into social settings, collaborative robots are becoming active participants in our everyday lives forging new frontiers in Human-Robot Interaction (HRI) that have the potential to change the way we interact with the world around us. As we navigate this era of increased collaboration between humans and machines, exploring the dynamics, challenges, and opportunities inherent in this symbiotic relationship becomes important.

While collaborative robots (cobots) are designed to assist humans, they still operate within highly predefined parameters that are constraining. For example, a robot will mostly stop or slow down when working in the periphery of humans, thus, limiting the impact of such collaboration [8]. In various sectors, fully autonomous cobots function within rigid frameworks, carrying out tasks alongside humans rather than engaging in genuine teamwork. As a result, while they enhance certain aspects of productivity and efficiency, their potential for seamless human-robot collaboration remains largely untapped [8]. In the case of human-human collaboration, communication is a crucial aspect that leads to successful teamwork and goal completion. Similarly, in human-robot teams, it is essential to have information sharing based on the human supervisory role and the robot's autonomy level. This can be achieved through interactions to synchronize goals, task states, and actions [6].

Virtual environments and simulations offer a valuable tool for comprehending the dynamics of interaction. They provide an intuitive platform for understanding the mechanics of how interactions would unfold. By leveraging virtual environments, adaptability can be significantly enhanced to cater to the specific requirements of the task space, the user involved, and the capabilities of the robot. These tools also provide the opportunity to evaluate interactions with virtual robots that are restricted by monetary and/or safety concerns in the real world [15].

Building upon our previous work [5], this paper presents a user study to compare *human-assisted* and *system-assisted* methods for human-robot collaboration. Figure 1 gives an overall understanding of the steps involved. The study investigates two collaboration frameworks: one where humans act as teleoperators and scene inspectors for robots (human-assisted, HA), and another where systems suggestions are taken as inputs, with robot teleoperation while humans make final decisions on areas of interest (system-assisted, SA). Objective and subjective measures are analyzed to elucidate factors influencing the development of intelligent and collaborative robots.

Two research questions guide this investigation:

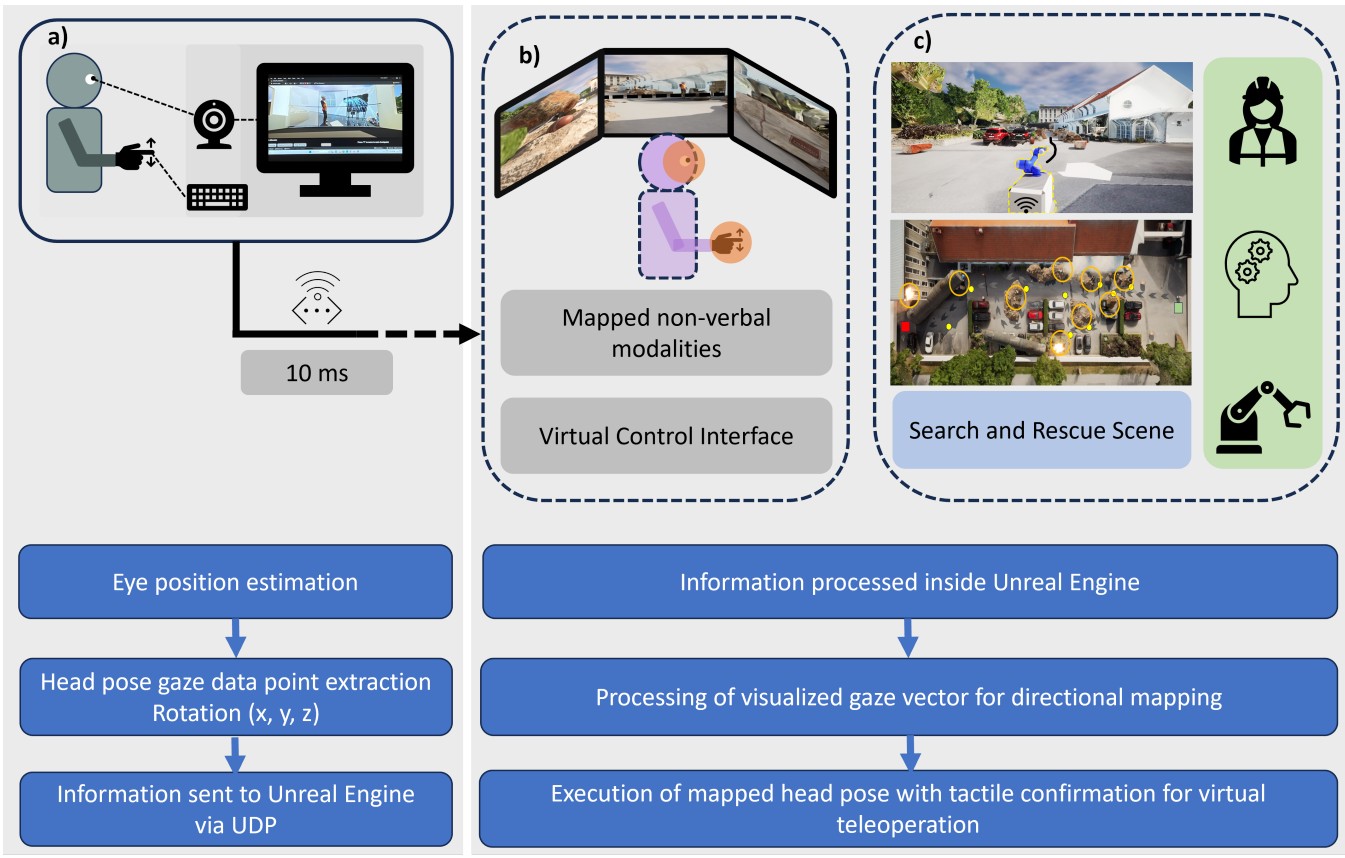

**Figure 1: An image of the different aspects of mixed-reality based human-in-the-loop robot control system. Here, a) the human operator's gaze and gesture modalities are taken as input using image recognition and tracking, b) the tracked modalities are mapped within the virtual control interface which in turn controls and monitors the agent in the system in human-assisted and system-assisted scenarios respectively, and c) search and rescue scene designed for assessing humans' intuition while performing different tasks.**

- "What level of human-robot collaboration is better at performing search operations in an unknown environment?"
- "How does a limited field of view affect goal execution?"

These questions aim to shed light on the effectiveness of human guidance in various tasks and the impact of cognitive load on goal execution within limited visual contexts.

## 2 RELATED WORK

### 2.1 Human-Robot Interaction (HRI) Frameworks and Communication Modalities

Human-robot interaction (HRI) encompasses a spectrum of interaction stages, as categorized by Onnasch et al. [9], including bounded autonomy, teleoperation, supervised autonomy, adaptive autonomy, and virtual symbiosis. However, the practical application of these stages often involves smooth transitions based on human roles and task demands, highlighting the importance of effective communication channels between humans and robots throughout these interactions. Researchers have investigated diverse communication modalities within HRI, surrounding two-way dialogue, natural language, multi-modal communication, and visual messages. While these modalities present rich interaction potentials, they often elevate cognitive workload and present hurdles to situational awareness. In response, discrete and sparse communication channels aimed at preserving human interpretability while strengthening decision-making precision have been suggested [8]. Gaze, identified as a natural means of interaction, has been leveraged in HRI, either as a primary input signal or in conjunction with other modalities [10]. However, gaze-only interfaces encounter challenges like the "Midas touch problem," where deciding when to select input becomes intricate due to the constant nature of gaze [14]. Consequently, separate confirmation mechanisms are necessitated to address these issues [11]. Techniques such as Eye & Head Dwell, Eye & Head Convergence, and Eye & Head Pointer have been investigated to enhance stability and efficacy in gaze-based interactions [11]. Moreover, head-supported gaze offers greater stability compared to gaze-only approaches [11]. Considering that humans utilize their bodies to attend to their environment or convey their attention to others, nonverbal cues like pointing or directing their

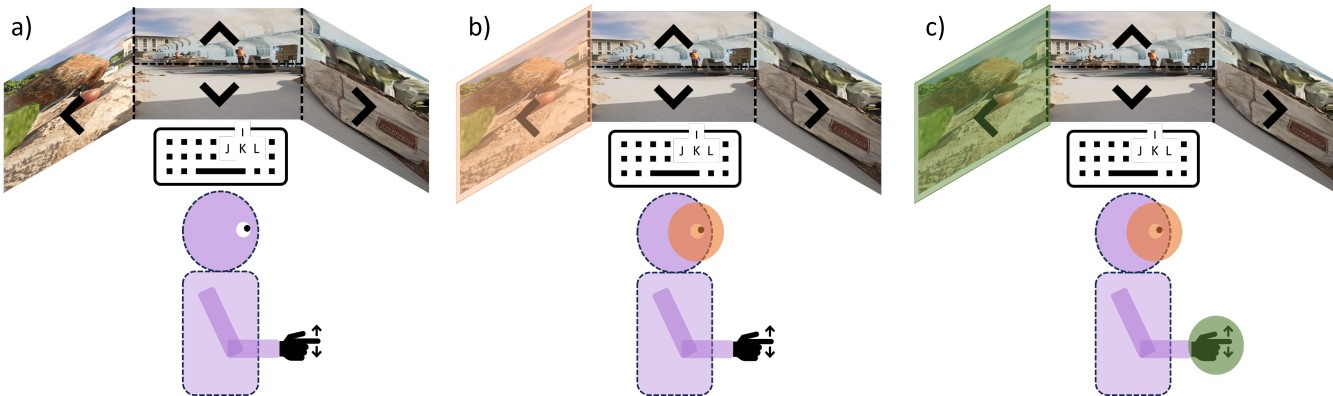

**Figure 2: Mapping of head pose and tactile confirmation**

head and eyes toward objects of interest emerge as natural candidates for further exploration in target selection and manipulation tasks within Extended Reality (XR) contexts [10].

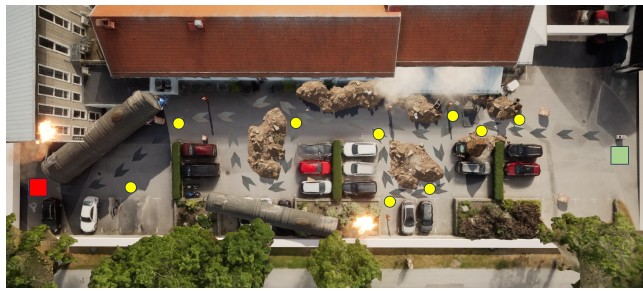

**Figure 3: Start (green box), goal (red box), and interaction target points (yellow box) of the robot during experiment in SA scenario**

## 2.2 Teleoperation Interfaces and Multi-modal Interaction Techniques

Teleoperation offers a bridge between human instinct and robotic capabilities [17]. Gesture-based teleoperation systems, utilizing devices like joysticks or motion-tracking devices, enable intuitive control methods for operators [17]. Immersive VR teleoperation interfaces replicate natural human motions, although they introduce complexities such as the need for specialized equipment [12]. This, when combined with head-supported gaze, can generate mapped motions in the interface [5]. Multi-modal interfaces play a crucial role in reducing cognitive workload and improving task performance in teleoperation scenarios [13]. These interfaces synchronize multiple modalities to enhance user immersion and awareness, contributing to more effective human-robot collaboration [13]. In the context of this work where humans need to perform faster searches, foveation methods also offer an interesting way to facilitate search mechanisms in cluttered and cognitively demanding environments [1].

In summary, research in the field has explored various communication modalities, teleoperation interfaces, multi-modal interaction

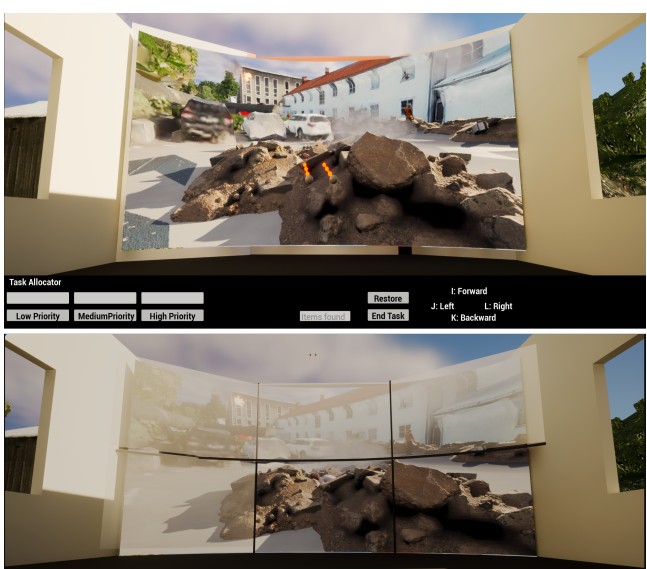

**Figure 4: FOVs of the participant in both the scenarios - HA scenario (top) and SA scenario (bottom).**

techniques, and foveation technologies to enhance human-robot collaboration across different interaction frameworks. These studies provide valuable insights for comparing the effectiveness of HA and SA approaches in HRI scenarios, as investigated in our current research.

## 3 EXPERIMENT

*3.1 Aim.* In this study, we aim to investigate the effectiveness and efficiency of a mixed reality-based system for improved human-robot collaboration, along with the underlying methods for user support in search and rescue situations involving otherwise autonomous systems.

**3.2 Test-bed environment.** To test in a search and rescue scenario we used an already existing 3D map [16] as the virtual environment and made modifications to create a simulated post-disaster scenario. Based on the interface design, similar 3D maps could be integrated at any stage to test other applications.

**3.3 Experimental Design and Workflow Steps.** Participants engaged in two sequential scenarios presented in random order. The two scenarios were designed with varying degrees of human intervention and decision-making. In the HA scenario, participants directed a simulated robot using eye gaze (left, right, up, down) and corresponding keyboard inputs (J, L, I, K) to traverse a search-and-rescue scene (figure 2), aiming to reach the end of a parking lot. This mode granted participants heightened control over task execution. This can be seen from the top part of figure 4.

In contrast, the SA scenario employed a Wizard-of-Oz technique to cluster identified areas of interest (AOIs) according to assigned importance levels (Low, Medium, High) for objects and humans in the scene. These AOIs could be seen as yellow point marks in figure 3, while green and red indicate start and end locations respectively. The robot autonomously navigated to these AOIs using foveation techniques, thus, reducing the cognitive load. Participants acted as final decision-makers, specifying their priority levels through the interface, utilizing similar importance categories (Low, Medium, High). This can be seen from the bottom part of figure 4.

**3.4 Task.** The overall goal of the task was to assess the scene and provide information regarding AOIs in a post-disaster scenario. The task for the participants was to count the points of interest they encountered and put corresponding priority markers for each of them. Participants were also provided with a small reference sheet before the experiment started to give a general idea of the importance of various objects in the scene.

**3.5 Participant Data, Recorded Information and Ethics.** Ethical considerations were taken into account before the experiment. As per the guidelines mentioned in [7] the experiment did not require an ethical review process from a committee. Participant demographics and recorded data, including log files, are anonymized, stored, and processed in line with the regulations of the university.

**3.6 After Experiment: Analysis.** We evaluated task completion time, task accuracy, and the number of identified humans to compare priorities between scenarios. These evaluations were complemented by workload analysis using NASA Task Load Index [4], system usability through the System Usability Scale [3], and subjective questionnaires to draw conclusive insights.

## 4 RESULTS

### 4.1 Participants

The total sample recruited for the user study consisted of 18 participants. There were 13 males, 4 females, and 1 Other with a mean age of 31.29 years (SD = 9.78) excluding 1 Other participant who refused to report their age. Out of the 18 participants, 7 had some level of vision impairment mostly corrected with eyeglasses. Since the task involved a search and rescue scenario, the use of multiple interfaces, and virtual scenarios, we were also concerned about the experience

of the participants in those aspects. Only 4 participants had experience in providing disaster relief. Participants also reported varying levels of experience across different domains: with robots (M = 2.72, SD = 1.7), with any form of virtual, augmented, or mixed reality system (M = 1.83, SD = 1.79), and with using controllers (M = 3.83, SD = 1.2). To eliminate any order and learning effects, half of the participants (N = 9) started the study with the HA scenario while the other half started with the SA scenario.

### 4.2 Task Timing

This is the first objective performance metric that we use to measure the performance of the participants in the two scenarios. Participants took an average time of 11m 17s (SD = 4m 34s) to complete the HA scenario and an average time of 3m 54s (SD = 53s) to complete the SA scenario. A paired two-sample t-test for two-tail significance of means shows that participants performed significantly better in the SA scenario (p < 0.001). This can be seen in figure 5.

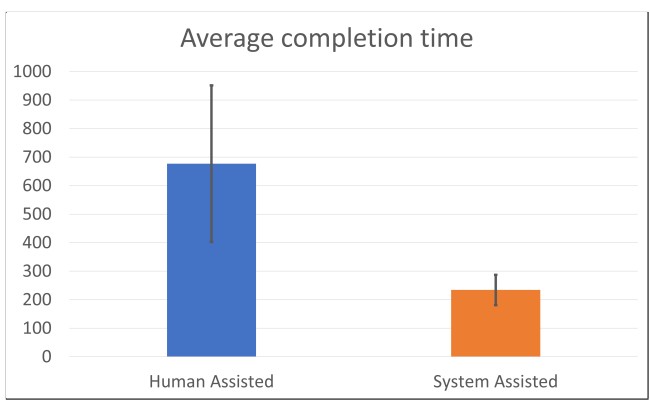

**Figure 5: Average completion time in seconds of participants in HA and SA scenarios**

### 4.3 Task Performance

The participants provided different priorities at different locations in the scenarios. Based on these, important locations and total instances were calculated for the identification of correct instances of the number of trapped humans present in the scene. There were 3 trapped humans in each scenario for the participants to locate during the task. In case of the HA scenario, out of N = 54 total instances, 20 (M = 1.11, SD = 0.76) were successfully identified. In case of the SA scenario, 47 (M = 2.6, SD = 0.7) instances were successfully identified. A paired two-sample t-test for two-tail significance of means shows that participants performed significantly better in the SA scenario (p < 0.001).

### 4.4 Workload

The participants answered the NASA TLX questionnaire after each scenario which helped measure the perceived workload for each scenario. The results show a high mean workload of 53.85 (SD = 18.07) in case of the HA scenario as compared to a mean workload of 33.41 (SD = 15.24) in the SA scenario with a paired two-sample t-test

for two-tail significance of means showing statistical significance (p < 0.001). This can be seen in figure 6.

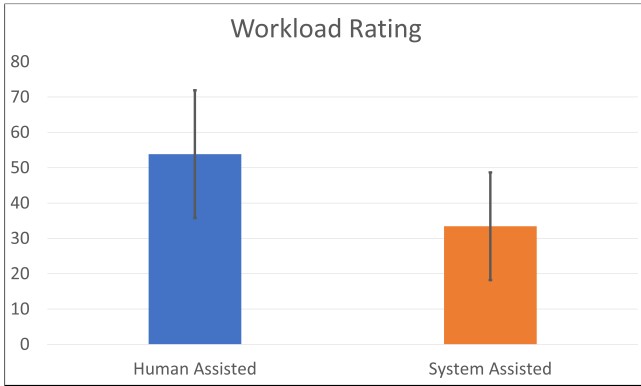

**Figure 6: Average workload experienced by participants in HA and SA scenarios**

## 4.5 System Usability

The system usability scale is a quick way to ascertain the usability of systems under scrutiny. Similar to the TLX questionnaire earlier, the participants answered 10 questions from the system usability questionnaire using a Likert scale (1 to 5) to indicate strong disagreement on the leftmost end (1) to strong agreement on the rightmost end (5). After calculating a single value from the responses to all 10 questions, the scores of the participants were averaged to arrive at the presented results. The participants reported an average usability of 58.61 (SD = 14.8) for the HA scenario and an average usability of 80.14 (SD = 16.3) for the SA scenario. Since the system usability score by itself does not represent a percentage, it needs to be normalized and converted to percentile to be interpreted correctly. According to [2], a system usability score of 68 marks the 50th percentile. A paired two-sample t-test for two-tail significance of means shows that participants preferred the SA scenario (p < 0.001). This can be seen in figure 7.

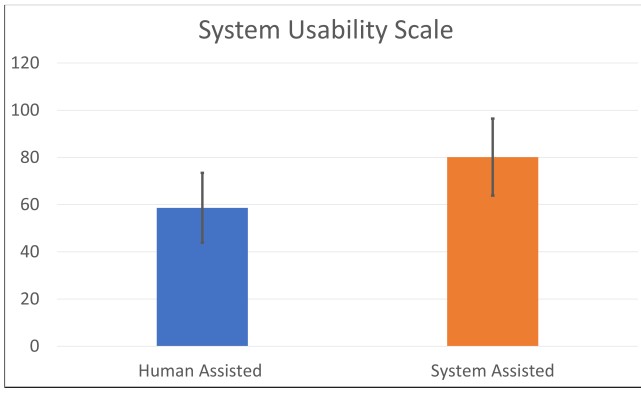

**Figure 7: Average system usability score reported by participants in HA and SA scenarios**

## 4.6 Subjective Assessment

The subjective assessment in the form of a questionnaire was presented to participants after the completion of each scenario followed by an end-of-experiment questionnaire. These questionnaires contained both long-answer form and five-point Likert scale-based questions. In the case of HA, the Likert scale-based questions were -

Q1 The non-verbal interactive interface helped me to provide assistance to the robot.
Q2 The non-verbal interface was intuitive and easy to use.
Q3 The robot accurately followed my guidance.
Q4 I am satisfied with the overall outcome of the search task.
Q5 My assistance contributed to the successful completion of the task.
Q6 Human assistance is beneficial for the robot in a search task in a cluttered environment.

The findings based on the responses are presented in the graph shown in figure 8.

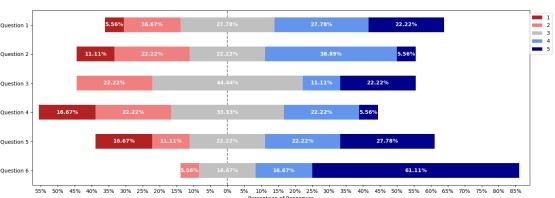

**Figure 8: Subjective Likert responses to HA scenario interview questions**

Similarly, in the case of SA, the Likert-based questions were -

Q1 I had a good experience with System assisted search for providing assistance to the robot
Q2 The foveated view field improved my experience in finding points of interest and importance in the scene
Q3 I trust the system's understanding of the scene to guide me to particular locations in the scene
Q4 I had a good experience with System assisted search and foveation for providing assistance to the robot in this case
Q5 I am satisfied with the overall outcome of the search task.
Q6 My assistance contributed to the successful completion of the task.
Q7 I am confident in the robot's ability to find the important locations.
Q8 Human assistance is beneficial for the robot in a search task in a cluttered environment.

The findings based on the responses are presented in the graph shown in the figure 9.

## 5 DISCUSSION

The results presented offer a comprehensive evaluation of the performance, workload, and usability of participants in two different scenarios: HA and SA. These scenarios were designed to assess the effectiveness and efficiency of systems in aiding users in identifying and locating trapped humans within a simulated environment. The

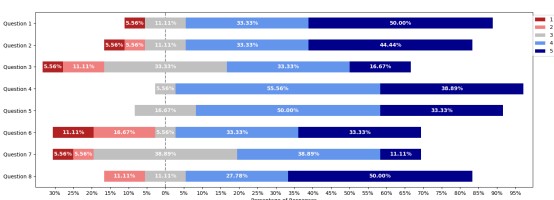

**Figure 9: Subjective Likert responses to SA scenario interview questions**

analysis and discussion below provide insights into the implications of these findings.

## 5.1 Task Timing

Participants completed the tasks significantly faster in the SA scenario compared to the HA scenario. The average time to complete the SA scenario was approximately one-third of the time taken to complete the HA scenario. This substantial reduction in task completion time suggests that the SA approach provides a more efficient means of accomplishing the task at hand.

## 5.2 Task Performance

In terms of task performance, participants demonstrated a higher success rate in identifying instances of trapped humans in the SA scenario compared to the HA scenario. The increased accuracy in identifying trapped humans indicates that the system provides valuable assistance to users, enhancing their ability to detect critical elements within the simulated environment.

## 5.3 Workload

The perceived workload reported by participants was significantly lower in the SA scenario compared to the HA scenario. This finding suggests that participants experienced reduced mental and physical demands when utilizing the SA approach. A lower perceived workload is desirable as it can lead to improved user satisfaction and overall performance.

## 5.4 System Usability

Participants rated the SA scenario as significantly more usable compared to the HA scenario. The higher system usability score indicates that participants found the system to be more intuitive, efficient, and satisfactory in assisting them with the task. The preference for the SA scenario underscores the importance of designing systems that are intuitive to use and supportive of user needs.

## 5.5 Subjective Assessment

The subjective assessment of teaming scenarios revealed valuable insights into the strengths and areas for improvement in both HA and SA scenarios. Participants shared detailed experiences and provided constructive feedback that can inform the refinement of systems for various human-robot collaborative tasks, particularly in scenarios involving emergency response and reconnaissance.

In the HA mode, participants demonstrated a preference for intuitive decision-making (ex - moving forward, looking around, size relates to danger), leveraging factors such as the likelihood of finding objects and the immediacy of danger to and around humans. However, challenges such as slow turning and limited peripheral vision were noted, highlighting the importance of improving physical interfaces and enhancing situational awareness. This can be also seen in the case of the Likert scale response, where participants generally had neutral or positive feedback about the non-verbal interface intuitiveness but there were mixed responses regarding the effectiveness of their assistance in guiding the robot accurately. Suggestions for improvement included implementing graphical interfaces for prioritizing items and enhancing navigation capabilities through features like independent perception control with depth feedback, joystick control, and sound cues. Participants also suggested to be provided with real-time feedback.

Conversely, in the SA mode, participants acknowledged the potential of automation in streamlining tasks and providing immediate feedback, particularly through features like foveation and object detection. However, concerns regarding the system's inability to highlight critical elements consistently and challenges related to foveation-induced loss of information were raised. Participants emphasized the importance of refining algorithms for scene perception and enhancing camera feeds to improve overall system performance.

We notice that although participants feel they can help the robot effectively, they don't fully trust the system's understanding of the environment. This suggests they're confident in their ability to assist practically but are unsure about how well the system comprehends the surroundings. This highlights the importance of aligning participants' perceptions of the robot's intent with its actual capabilities to foster trust and collaboration.

Furthermore, discussions surrounding the handover of control between human operators and the robot highlighted the necessity of clear protocols and established cooperation practices. While participants expressed willingness to delegate control under certain conditions, such as when the operator possesses superior situational awareness or familiarity with the task, concerns regarding potential conflicts and the need for a hierarchical command structure were evident.

## 6 CONCLUSION AND FUTURE DIRECTION

In this study, we set out to investigate the effectiveness and efficiency of mixed-reality-based systems in assisting operators during human-robot collaboration scenarios. Experiments with participants in a virtual search and rescue environment helped us explore this using multimodal interaction techniques. Participants demonstrated improved task performance, reduced workload, and higher usability ratings when utilizing SA methods compared to HA ones. In both cases, the results emphasize the complex interplay between human intuition and automated assistance in collaboration scenarios. Subjective assessments highlighted the importance of intuitive interfaces, real-time feedback, and clear protocols for effective collaboration between human operators and robotic systems. By addressing the identified challenges and incorporating user feedback, future developments in human-robot teaming can

enhance operational effectiveness and promote seamless collaboration between human operators and robotic systems, ultimately advancing capabilities in domains such as emergency response and reconnaissance.

## 7 ACKNOWLEDGEMENT

We would like to thank Jacek Malec and Björn Olofsson for their support during the work. This work was supported by ELLIIT - the Excellence Center at Linköping University and Lund University for Information Technology. The 3D environment employed in this research is provided by the WASP Research Arena for Public Safety (WARA-PS). The Wallenberg Artificial Intelligence, Autonomous Systems, and Software Program (WASP) funded this initiative through the Knut and Alice Wallenberg Foundation.

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
