# OpenReview forum: "Towards Understanding the Role of Humans in Collaborative Tasks"
_humanrobotinteraction.org/HRI/2024/Workshop/VAM-HRI — VAM-HRI 2024 Oral_

### Official Review · Reviewer_fAgX · 2024-02-20
**Accept**

**Rating:** 7
**Confidence:** 4

**Review:**

The paper explores human-robot collaboration (HRC) within a search and rescue context, comparing human-assisted and system-assisted approaches using virtual and mixed-reality interfaces. It investigates the dynamics, effectiveness, and user experiences of these collaboration modes, demonstrating that system-assisted methods significantly improve task performance, reduce workload, and are preferred by users for their efficiency and usability.

## Strengths:
- **Empirical Approach:** Utilizes a comparative study with objective and subjective metrics to evaluate collaboration modes.
- **Technological Integration:** Leverages advanced mixed-reality interfaces for immersive and effective HRC.
- **Valuable Insights:** Offers substantial evidence on the benefits of system-assisted collaboration, including efficiency and user satisfaction.

## Weaknesses:
- **Limited Scope:** Focuses on a specific application area (search and rescue), which may restrict the generalizability of the findings.
- **Sample Size:** Relatively small participant sample could impact the robustness of the study's conclusions.

## Recommendations for Improvement:
- **Expand Research Scope:** Broaden the study to include various HRC contexts for greater applicability.
- **Increase Participant Diversity:** Enlarge and diversify the participant pool to enhance the study's generalizability.
- **Further Technological Exploration:** Explore additional mixed-reality technologies and interfaces to identify other potential benefits and drawbacks in HRC scenarios.

In summary, I think this paper is a great fit for VAM-HRI, and I recommend acceptance.

---

### Official Review · Reviewer_T5AV · 2024-02-23
**Accept**

**Rating:** 7
**Confidence:** 4

**Review:**

This paper compares a human-assisted and a system-assisted method for human-robot collaboration during a search and rescue task. The human-assisted interface moves the camera based on a user’s eye gaze and keyboard input, while the system-assisted method autonomously navigates an environment while areas of interest are marked by yellow to direct a user’s attention to help them make decisions using necessary information. A within-subjects study was conducted with 18 participants, with results pointing to the system-assisted design as being a superior interface than the human-assisted one.

Strengths:

- The paper utilizes a fairly realistic simulation that showcases the potential for virtual environments as test beds for search and rescue research.

- The system implements multiple components of user interaction that provide interesting avenues of work. For example, rather than having users direct a robot with gaze, perhaps the robot directs the user themselves.

Areas of Improvement:

- In future work, this study could be strengthened by incorporating an industry standard user interface as a baseline condition. For example, I would be curious to see how the time to complete the task by controlling the robot with a joystick controller and camera stream would compare to the SA system.

- The discussion could provide a more in depth analysis of the results. For example, rather than rephrasing Section 4 in Section 5, perhaps the paper could provide more of a synthesis of the results. Are there important insights as to why these results might have been achieved? The SA system seemed to be significantly better than the HA system, are there components or features that can be applied to other interfaces?

- It is slightly unclear how conducting this study in VR provides enhancements that cannot be achieved by a curved monitor. Were participants provided a stereoscopic view of the scene?

Overall, this paper is a great start towards evaluating alternate VR interfaces for human-robot collaboration during search and rescue. I look forward to seeing the next stages of this work.

---

### Decision · Program_Chairs · 2024-02-26

Accept (Oral)